# Distinct beta frequencies reflect categorical decisions

Elie Rassi[1,2], Yi Zhang[3], Germán Mendoza[4], Juan Carlos Méndez[5,6], Hugo Merchant ®[4] ✉ & Saskia Haegens ®[1,3,7]

Based on prior findings of content-specific beta synchronization in working memory and decision making, we hypothesized that beta oscillations support the (re-)activation of cortical representations by mediating neural ensemble formation. We found that beta activity in monkey dorsolateral prefrontal cortex (dlPFC) and pre-supplementary motor area (preSMA) reflects the content of a stimulus in relation to the task context, regardless of its objective properties. In duration- and distance-categorization tasks, we changed the boundary between categories from one block of trials to the next. We found that two distinct beta-band frequencies were consistently associated with the two relative categories, with activity in these bands predicting the animals' responses. We characterized beta at these frequencies as transient bursts, and showed that dlPFC and preSMA are connected via these distinct frequency channels. These results support the role of beta in forming neural ensembles, and further show that such ensembles synchronize at different beta frequencies.

Categorization is the ability to group objects or events that share certain features. Depending on the context, the same feature could place an object or event in different categories[1]. For example, in categorizing "big" vs. "small", a 10-kg cat is considered a big cat while a 10-kg elephant is considered a small elephant. Further, category boundaries tend to be flexible, and new arbitrary categories can be learned[2–4]. For example, if one learns that 10 kg is, in fact, below average for a cat, one could shift their category boundary and consider it small.

Neural oscillations, which are ubiquitous across species, are widely believed to support brain function[5–7]. Indeed, oscillations are thought to play a key role in supporting categorization[8–10], but little is known about the oscillatory dynamics that allow the brain to achieve such context-dependent and flexible categorization. Is context-dependent "small" signaled in the same way, regardless of absolute size? Once a new category boundary is learned, are the newly learned categories signaled in the same way as the old ones? Here, we used a context-dependent categorization task, where the boundary between two categories varied on different blocks, meaning the same stimulus could be classified as belonging to different categories depending on the task context[11]. In each block, Rhesus monkeys categorized eight intervals (temporal task) or eight distances (spatial task) as short or long according to a criterion defined at the start of the block.

Local field potentials (LFPs) and single-cell activity were simultaneously recorded in the dorsolateral prefrontal cortex (dlPFC) and pre-supplementary motor area (preSMA). dlPFC is known to play a central role in categorization[12] and is part of the magnitude system for time, space, and quantity[13]. It is deeply connected with preSMA, which is known to be a major node in the time processing network[14], and contains cells that have been shown to encode the boundary between categories[11].

We focused on beta oscillations (~15–35 Hz) and hypothesized that beta activity supports the activation and reactivation of cortical

[1]Donders Institute for Brain, Cognition and Behaviour, Radboud University, Nijmegen, The Netherlands. [2]Department of Psychology, Centre for Cognitive Neuroscience, Paris-Lodron-University of Salzburg, Salzburg, Austria. [3]Department of Psychiatry, Columbia University, New York, NY, USA. [4]Instituto de Neurobiología, UNAM, Campus Juriquilla, Queretaro, Mexico. [5]Department of Experimental Psychology, University of Oxford, Oxford, UK. [6]College of Medicine and Health, University of Exeter, Exeter, UK. [7]Division of Systems Neuroscience, New York State Psychiatric Institute, New York, NY, USA. ✉e-mail: hugomerchant@unam.mx

representations by mediating neural ensemble formation within and between brain regions[15]. Beta has been implicated in top-down processing and preservation of the current brain state[16]. Beta synchronization can be content-specific during endogenous information processing[8,17,18], typically characterized as short-lived and reflecting flexible network dynamics[19,20]. There is evidence that beta is highly context-specific, flexibly carrying information relevant to the task at hand[21–23]. This makes beta-band synchrony a candidate mechanism involved in context-dependent categorization. The separate representation of different categories by different neural ensembles could be mediated by beta synchrony at distinct frequencies or with varying power. Indeed, we found that consistently and across all versions of the task, two distinct beta-band frequencies reflected the two contextually-defined categorical decisions.

## Results

Two monkeys were trained on both temporal (interval) and spatial (distance) categorization tasks, where they categorized the duration of a time interval or the distance between two vertical bars as either "short" or "long" (Fig. 1)[11]. A trial started when the animal fixated centrally and maintained a cursor in a central circle. After a 500-ms pre-stimulus delay, two parallel bars were visually presented twice, separated by a delay (Fig. 1a). In the temporal task, the delay varied, and the distance between bars remained constant, while in the spatial task, the distance between bars varied and the delay duration remained constant (Fig. 1b). Although the distance between bars varied between trials in the spatial task, it was the same within a trial in both task versions. After the stimuli were presented, there was a fixed decision delay of 500 ms (monkey 1) or 1000 ms (monkey 2), on which we focused our analyses (Fig. 1a). Crucially, during the decision delay, the monkey would not yet know the response mapping (i.e., which cursor movement corresponded to which category) as this varied on each trial, thereby precluding motor-related preparatory activity.

In every recording session, monkeys categorized four to six different blocks of stimuli (three temporal: T1–T3; three spatial: S1–S3), each containing eight different time intervals or distances (four short and four long; Fig. 1b). Each block started with 24 training trials, in which only the shortest and longest stimuli (relative to that particular block) were presented in random order, such that the monkeys learned an implicit value for the boundary between the two categories for that block. After this training phase, all eight stimuli were presented in random order across 96 testing trials, which we used for our analyses. Because of this design, every time a new block was tested, the boundary was shifted, such that the same stimulus could be categorized as long in one block but short in another (e.g., a 450-ms interval would be long in T1 but short in T2; see overlaps in Fig. 1b). A complete description can be found in the Methods section. On average, monkey 1 performed the tasks with 69% accuracy and Monkey 2 with 72% accuracy.

### Beta activity increased in frequency from prestimulus to decision delay

We analyzed LFPs from 217 experimental blocks (1 electrode per block; 199 from monkey 1; 18 from monkey 2) in preSMA and 199 experimental blocks (one electrode per block; all from monkey 1) in dlPFC (Fig. 1c). In both preSMA and dlPFC, we observed oscillatory activity predominantly in the beta frequency band during the prestimulus delay (500 ms preceding onset of the first stimulus). During the decision delay, beta remained the dominant oscillation, but its power was mostly suppressed in both sites compared to the prestimulus delay (Fig. 2a–c; monkey 1 dlPFC: significant frequency range 15–27 Hz, cluster-corrected $p = 0.0013$; monkey 1 preSMA: significant frequency range 15–27.5 Hz, cluster-corrected $p = 1e-6$; monkey 2 preSMA: significant frequency range: 15–27, cluster-corrected $p = 0.0047$). Additionally, for monkey 1, beta power was enhanced in the higher beta frequencies in both sites during the decision delay compared to the prestimulus delay (Fig. 2a, b; dlPFC: significant frequency range

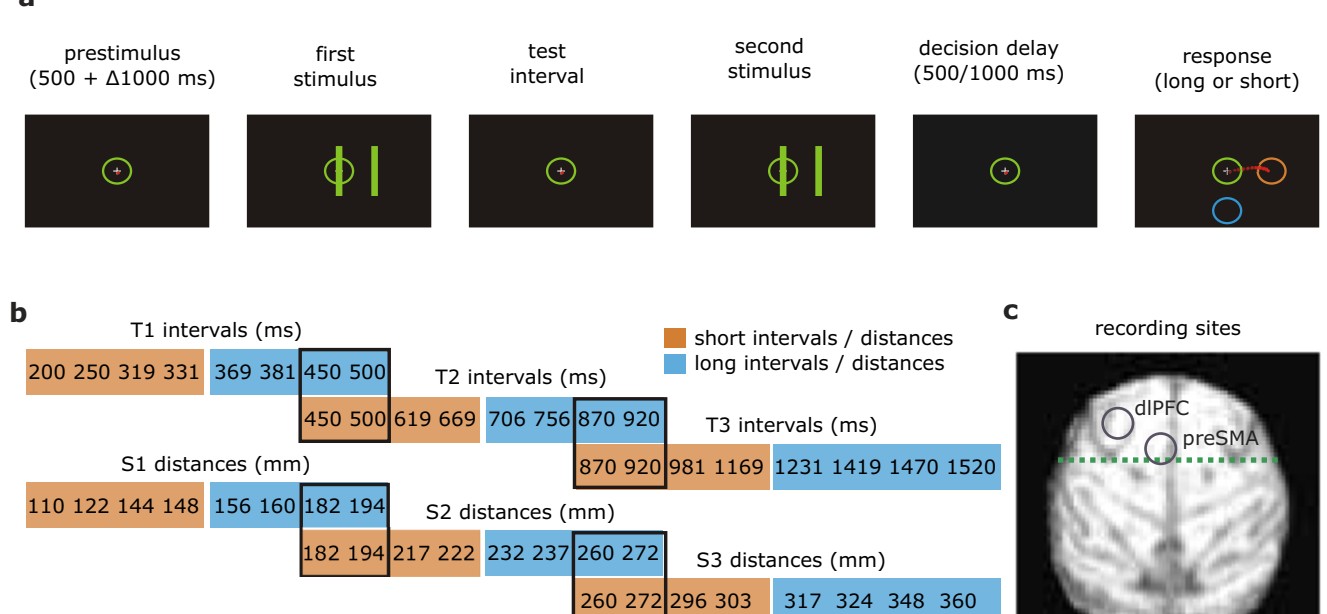

**a**

| prestimulus (500 + Δ1000 ms) | first stimulus | test interval | second stimulus | decision delay (500/1000 ms) | response (long or short) |

**b**

T1 intervals (ms)
| 200 | 250 | 319 | 331 | 369 | 381 | 450 | 500 |

T2 intervals (ms)
| 450 | 500 | 619 | 669 | 706 | 756 | 870 | 920 |

T3 intervals (ms)
| 870 | 920 | 981 | 1169 | 1231 | 1419 | 1470 | 1520 |

S1 distances (mm)
| 110 | 122 | 144 | 148 | 156 | 160 | 182 | 194 |

S2 distances (mm)
| 182 | 194 | 217 | 222 | 232 | 237 | 260 | 272 |

S3 distances (mm)
| 260 | 272 | 296 | 303 | 317 | 324 | 348 | 360 |

■ short intervals / distances
■ long intervals / distances

**c** recording sites

dlPFC
preSMA

**Fig. 1 | Trial design, stimuli, and recording sites. a** After a pre-stimulus delay, two visual stimuli were shown, separated by an interval. Shortly after the disappearance of the second stimulus, the animal would indicate whether the duration of the interval (temporal tasks) or the distance between the bars (spatial tasks) was "long" or "short". Note that during the decision delay, the monkey could not yet indicate their decision and did not yet know which cursor movement corresponded to which decision. The movement-to-decision mapping was randomized in each trial. **b** Stimuli (intervals and distances) were used across the six tasks. **c** Locations of the recording sites: dlPFC and preSMA. Reproduced with permission from Mendoza, G., Méndez, J.C., Pérez, O. et al. Neural basis for categorical boundaries in the primate pre-SMA during relative categorization of time intervals. Nat Commun 9, 1098 (2018). https://doi.org/10.1038/s41467-018-03482-8.

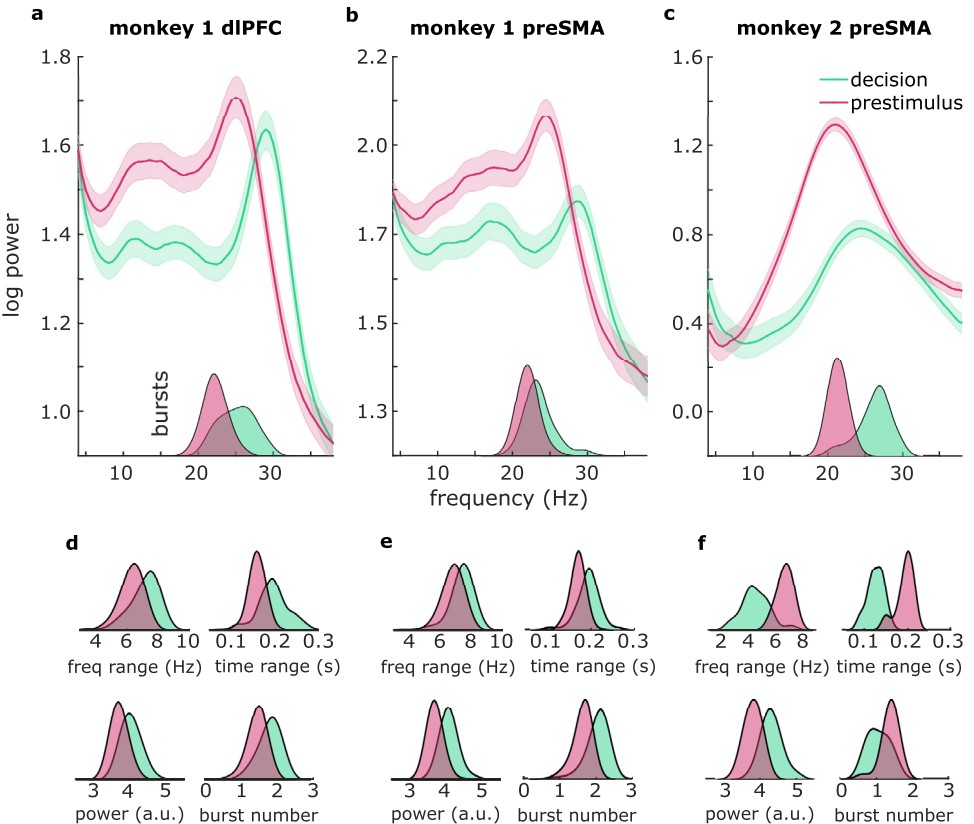

**Fig. 2 | Frequency shift and distinct burst profiles between prestimulus and decision intervals. a** Spectral power (line graphs) and burst frequency (density plots in insets) during decision (green) and prestimulus delays (pink) in monkey 1 dlPFC, **b** monkey 1 preSMA, and **c** monkey 2 preSMA. Shaded regions around the line graphs represent the standard error of the mean. **d** Density plots represent burst frequency range, time range, power, and number in monkey 1 dlPFC, **e** monkey 1 preSMA, and **f** monkey 2 preSMA. Source data are provided as Source Data files.

28–35 Hz, cluster-corrected $p = 0.0088$; preSMA: significant frequency range 28.5–35 Hz, cluster-corrected $p = 0.0096$). This simultaneous power decrease in the lower beta band and increase in the higher beta band was indicative of an upwards shift in beta peak frequency between prestimulus and decision delays. For monkey 1, the peak frequency shifted from 25 to 29 Hz in dlPFC (Fig. 2a; Table S1; $t(198) = 23.3$, $p < 1e-12$) and from 24 to 27 Hz in preSMA (Fig. 2b; Table S2; $t(198) = 12.7$, $p < 1e-12$), for monkey 2 the peak frequency shifted from 21 to 24 Hz in preSMA (Fig. 2c; Table S3; $t(17) = 6.8$; $p = 3e-6$;).

Since beta oscillations have been characterized as transient bursts of high-amplitude activity[19,20], we zoomed in on the beta band bursting profile and found an analogous frequency shift: compared to the prestimulus delay, the peak frequency of beta bursts significantly increased during the decision delay for both monkeys and both regions (Fig. 2a–c insets; see Tables S1–3 for statistics). In addition, bursts had distinct temporal and spectral profiles during the two periods (Fig. 2d–f; see Tables S1–3 for statistics). For monkey 1, bursts were more prominent in both regions during the decision delay: they were higher in number and had larger peak amplitudes, time ranges, and frequency ranges (see Fig. 2d and Table S1 for dlPFC, and Fig. 2e and Table S2 for preSMA). For monkey 2, bursts were higher in number and had larger time and frequency ranges in the prestimulus delay but higher peak power during the decision delay (Fig. 2f and Table S3).

**Two distinct beta-band frequencies reflected the two relative categorical decisions across tasks and recording sites**
Next, we focused on the decision delay, in which the monkeys would form a decision and hold it in memory. We contrasted the LFP power spectra for trials in which the monkeys correctly categorized stimuli as "short" vs. "long". When pooling across all versions of the task, we found a consistent frequency shift of around 2 Hz in the beta range, such that beta on "short" trials was 2 Hz faster than beta on "long" trials. This was the case for both recording sites (Fig. 3a, b) and both monkeys (Monkey 1 dlPFC: $t(198) = 11.0$, $p < 1e-15$, AUROC = 0.751; Monkey 1 preSMA: $t(198) = 8.5$, $p = 5e-15$, AUROC = 0.724; Monkey 2 preSMA: $t(17) = 3.5$, $p = 0.003$, AUROC = 0.660; see Tables S4 and 5 for statistics on individual task versions). This frequency separation between "long" and "short" categories was also evident when looking at the bursts' peak frequencies. (Fig. 3a, b insets; Tables S6–8). Given that we found this main result for both monkeys and that our next planned analyses relied on sub-sampling experimental trials, we pooled the data across both animals at this point to maintain statistical power.

To assess the behavioral relevance of the aforementioned frequency shift, we analyzed trials where the stimuli were incorrectly categorized and found that the shift was reversed (Fig. 3c, d), such that the beta rhythm on incorrect "short" trials (i.e., trials in which the correct category would be "long") was faster than the beta rhythm on incorrect "long" trials (Fig. 3c, d; dlPFC: $t(198) = 6.8$, $p = 1e-10$; preSMA: $t(216) = 5.7$, $p = 4e-8$). Similarly, the effect was reversed when looking at burst frequency (Fig. 3c, d insets; dlPFC: $t(198) = -6.6$, $p = 5e-10$; preSMA: $t(216) = -5.5$, $p = 2e-7$). This shows that beta frequency reflected the categorical decision that was held in memory, regardless of whether it was correct or not.

We found the same frequency shift in both recording sites when looking at each task individually as well, such that beta on "short" trials was always faster than on "long" trials in all tasks (Tables S4 and S5), regardless of the absolute magnitude of the stimuli in those tasks. As an example to illustrate this point, beta during "very short" trials of T3

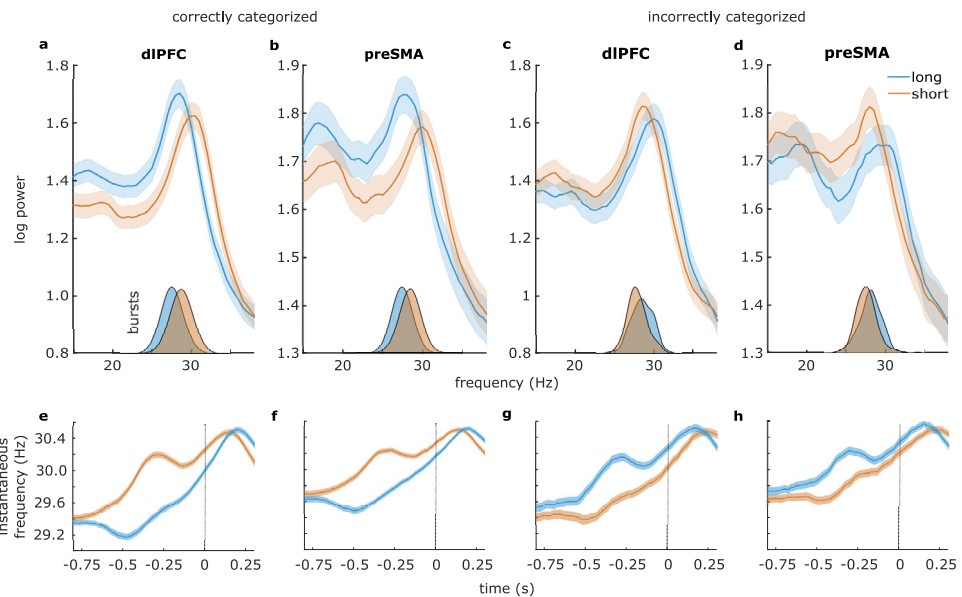

**Fig. 3 | Beta peak frequency reflected the categorical decision during the decision delay. a–d** Power spectra for "long" stimulus (blue) vs. "short" stimulus trials (orange) during trials with correct (**a**, **b**) and incorrect (**c**, **d**) responses. Insets: burst peak frequencies. **e–h** Instantaneous frequency time courses. Time zero represents the end of the decision delay. Shaded regions around all line graphs represent the standard error of the mean. Source data are provided as Source Data files.

was faster than beta during "very long" trials of T2 (unpaired *t*-tests; dlPFC: $t(77) = 6.1$, $p = 5e−8$; preSMA: $t(86) = 3.4$, $p = 9e−4$), despite these intervals being exactly the same (see overlapping stimuli outlined in Fig. 1, and see Tables S9 and S10 for statistics on each overlapping stimulus). The same was true when comparing the identical "very short" T2 trials with "very long" T1 trials (dlPFC: $t(99) = 2.4$, $p = 0.019$; preSMA: t(99) = 2.6, p = .010). Critically, beta during "very short" trials was not faster nor slower than beta during the "less short" trials (nor was beta during "very long" trials different from "less long" trials), as these stimuli were within the same category (2-way ANOVA; dlPFC: main effect "long vs. short" $F = 136$, $p < 1e−15$, no main effect "very vs. less", no interaction; preSMA: main effect "long vs. short" $F = 80$, $p < 1e−15$, no main effect "very vs. less", no interaction). In sum, beta frequency reflected the categorical decision, which was relative to each task, rather than the absolute magnitude of the stimulus.

To investigate the time course of these frequency shifts, we then analyzed the instantaneous frequency[24] in the beta range. We again found a clear separation between beta frequencies on "short" vs. "long" trials throughout the decision window (Fig. 3e, f; dlPFC: cluster-corrected $p < 1e−12$; preSMA: cluster-corrected $p < 1e−12$), with the differences peaking at around 350 ms (dlPFC) and 360 ms (preSMA) prior to the onset of the response screen (Fig. 3e, f). On incorrectly categorized trials, the instantaneous frequency time-course was near-identical to that of the correctly categorized trials, but the directionality of the frequency difference was again reversed (Fig. 3g, h; dlPFC: cluster-corrected $p < 1e−12$; preSMA: cluster-corrected $p < 1e−12$). That is, "long" trials incorrectly categorized as "short" showed a higher beta frequency than "short" trials incorrectly categorized as "long" throughout the decision window (Fig. 3g, h), again highlighting the behavioral relevance of the frequency separation.

Furthermore, the two trial types had distinct burst profiles consistent across the two recording sites (Tables S6 and S7). The peak frequencies at which beta bursts occurred showed the expected pattern of results; that is, bursts during "short" trials had a higher peak frequency compared to "long" trials (Fig. 3a, b insets; dlPFC: $t(198) = 10.8$, $p < 1e−12$; preSMA: $t(216) = 9.9$, $p < 1e−12$), and the effect was again reversed on incorrect trials (Fig. 3c, d insets; dlPFC: $t(198) = 6.6$, $p = 4.5e−10$; preSMA: $t(216) = 5.4$, $p = 1.6e−7$). Moreover,

bursts during "short" trials were lower in number and had smaller amplitudes and time and frequency spans compared to bursts on "long" trials (see Tables S6 and S7 for statistics). Overall, multiple analysis approaches confirmed that two distinct beta frequencies with distinct spectro-temporal profiles reflected the two relative categorical decisions, regardless of the objective stimulus properties or of whether the decisions were correct or not.

## Inter-areal connectivity and spike-field coherence at the same distinct frequencies also reflected the relative categories

As our theoretical framework postulated that inter-areal beta connectivity contributes to the activation of local ensembles[15], we then analyzed the connectivity between our two recording sites (preSMA and dlPFC). Note that these simultaneous recordings were only available in one animal, so the following results should be treated with caution. We found a frequency shift analogous to that reported above: coherence between the two regions during the decision delay of "short" trials peaked at a higher frequency compared to "long" trials (Fig. 4a; $t(198) = 5.4$, $p = 1e−7$). On incorrectly categorized trials, the frequency shift was reversed, such that coherence on "long" trials incorrectly categorized as "short" peaked at a higher frequency than "short" trials incorrectly categorized as "long" (Fig. S4a; $t(163) = −3.0$, $p = 0.003$). We then investigated the directionality of this connectivity effect with nonparametric Granger causality. We again found the same beta frequency shift, but this effect depended on the directionality of Granger causality (Fig. 4c, d; 2-way ANOVA; main effect "direction" $F(1, 792) = 8$, $p = 0.005$; main effect "short vs. long" $F(1,792) = 4.6$, $p = 0.033$, interaction $F(1,792) = 9.3$, $p = 0.002$), with the beta frequency shift occurring in the direction from dlPFC to preSMA ($t(198) = 3.7$, $p = 2.6e−4$), but not in the direction from preSMA to dlPFC ($t(198) = 0.54$, $p = 0.59$, ns; difference in peak shifts between the two directions [dlPFC → preSMA vs. preSMA → dlPFC]: $t(183) = 2.9$, $p = 0.004$). On incorrectly categorized trials, the frequency shift was reversed, such that dlPFC to preSMA Granger causality on "long" trials incorrectly categorized as "short" peaked at a higher frequency than "short" trials incorrectly categorized as "long" (Fig. S4b; $t(169) = −3.8$, $p = 2e−4$). In the direction of preSMA to dlPFC, there were no differences in Granger causality peak frequencies between the two trial

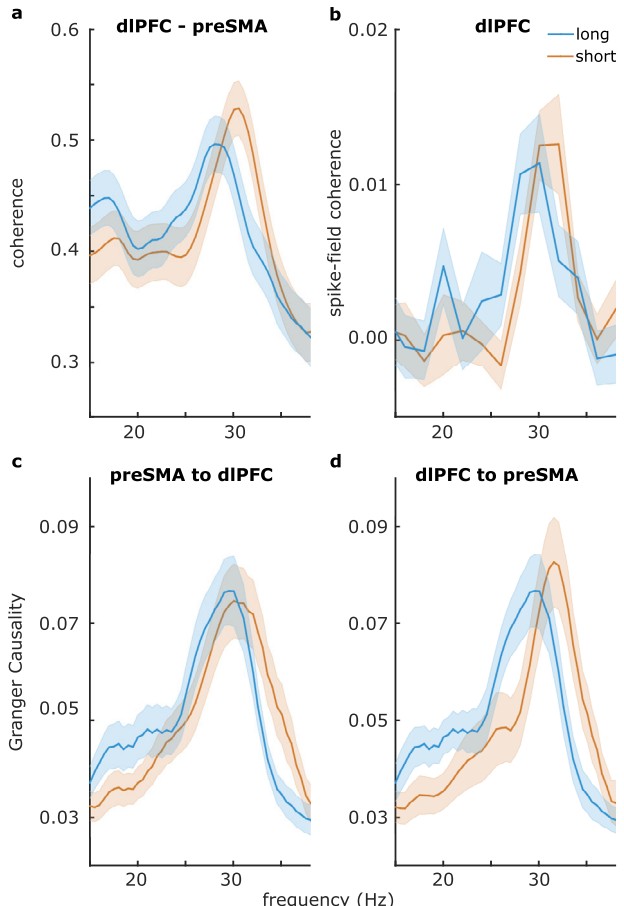

**Fig. 4 | Peak frequencies of between-region connectivity and spike-field coherence in dlPFC reflected the categorical decision during the decision delay. a** Coherence between preSMA and dlPFC. The peak frequency of coherence reflected the categorical decision. **b** Spike-field coherence in dlPFC between short-selective neurons and the LFP during short-categorized trials and between long-selective neurons and the LFP during long-categorized trials: the peak frequency reflected the categorical decision. **c, d** Granger causality between preSMA and dlPFC. The peak frequency of dlPFC to preSMA Granger causality, but not preSMA to dlPFC Granger causality, reflected the categorical decision. Shaded regions around the line graphs represent the standard error of the mean. Source data are provided as Source Data files.

types on incorrectly categorized trials (Fig. S4c; $t(169) = −0.7$, $p = 0.50$, ns). In sum, a beta frequency drive from dlPFC to preSMA was separated into two distinct beta frequency channels, which reflected the categorical decisions, not the stimuli.

Finally, we analyzed the spike-field coherence in both regions using pairwise phase consistency, a measure of how consistently neuronal spikes occurred at a particular phase and frequency. After sorting neurons based on whether they responded to "short" or "long" categories (see Methods section for details), we calculated spike-field coherence separately for short-selective neurons (preSMA $n = 478$, dlPFC $n = 445$) with "short" trials, and long-selective neurons (preSMA $n = 391$, dlPFC $n = 329$) with "long" trials. Note that there were no significant differences in the firing rates of these "short" vs. "long" cells (both $t < 1$, both $p > 0.3$). Again, here we found a beta frequency shift in dlPFC (Fig. 4b; $t(441) = 2.3$, $p = 0.023$), but not in preSMA ($t(192) = −0.6$, $p = 0.56$, ns). This meant that the two groups of neurons in dlPFC spiked in synchrony with the two beta frequency bands. In other words, short-selective neurons cohered with the beta at the frequency reflecting a "short" decision, and long-selective neurons cohered with the beta at the frequency reflecting a "long" decision. We found that this in dlPFC is in line with the dlPFC to preSMA direction of

connectivity we reported above. Overall, the frequency shift reported throughout this study is visible in both preSMA and dlPFC, but it seems to be driven by dlPFC and visible in dlPFC spike-field interactions. However, we cannot draw strong conclusions from the above results relating to connectivity and spike-field coherence, as these results rely on data from one animal.

## Discussion

In a temporal and spatial categorization task, we changed the categorical boundary within sessions such that a stimulus that belonged to one of two categories during one block of trials could belong to the other category during the next block. Two monkeys implicitly learned the categorical boundary at the start of each session and, depending on the task categorized either a time interval or the distance between two bars as being either short or long. Across all the different versions of the task, we found that two distinct beta-band frequencies were consistently associated with the same two relative categories during a decision delay in both preSMA and dlPFC. The two frequency channels had distinct spectral, temporal, connectivity, and spike-field coherence profiles. The frequencies were specifically associated with the decisions to categorize "short" vs. "long" relative to each block. This frequency specificity was evident despite the change in the type of stimuli (time intervals or distances) and despite stimuli with identical magnitudes falling into different categories depending on the context-defined boundary.

While beta oscillations are traditionally associated with sensorimotor functions[25], the current experiment was carefully designed to preclude any sensorimotor effects: the monkeys did not yet know which movement to make to indicate their decision during the delay that we analyzed here. Only after the decision delay was that information revealed, and it was random on each trial. Therefore, the observed beta dynamics reflected the categorical decision held in mind during the delay and not preparatory motor activity.

The beta frequencies were associated with the contextually-defined categorical decisions but not with the absolute magnitudes of the stimuli. In the cases where the same stimulus would fall into one category in one block and into the other category in a subsequent block, the beta signal still provided a read-out of the category relative to the block. In other words, a beta frequency reflected a relative and abstract concept of "short" or "long"[26]. This was also true at a higher level of abstraction, as "short" and "long" could refer to the duration of a time interval or to the distance between two visual stimuli. Even when the monkeys made errors, the beta signal still provided a read-out of the monkey's incorrect decision rather than the actual category of the stimulus. In other words, the monkey's behavioral response could be predicted from the beta signal.

The frequency separation was evident in multiple aspects of the data. We characterized beta activity as transient burst events and found that the bursts corresponding to the two relative categories differed in frequency. While this was the most salient feature, the number, power, timing, and time and frequency spans of the bursts were also linked with the categorical decisions. The frequency separation started at the beginning of the decision delay and peaked toward the middle before disappearing at the end of the delay, which is when the monkeys were allowed to indicate their decision.

In line with the proposed role of the prefrontal cortex in abstract categorization and higher-order cognitive functions[3,8,10], we found a prominent beta frequency shift in dlPFC. Spike-field coherence analyses revealed that the relevant neurons synchronized with the relevant oscillatory frequencies in dlPFC. Using a neural category-boundary signal, single cells in preSMA and dlPFC encode the categorical decision, with different sets of neurons activating for short and long decisions during the delay[11]. Here we found that each of these sets of dlPFC category-selective neurons synchronized with the beta rhythm at the respective category-selective beta frequency: the set of

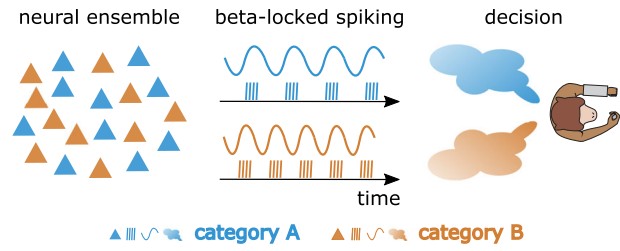

**Fig. 5 | Schematic of neural ensembles selectively signaling categorical decisions via distinct beta frequency channels.** Left: Two overlapping neural ensembles are selective for two different categories. Triangle icons represent neurons. Middle: Spiking activity of the two neural ensembles is coherent with two different beta frequencies; category information is transmitted downstream via these distinct beta frequency channels. Vertical lines represent spike trains. Waves represent beta rhythms. Right: The two beta rhythms at distinct frequencies provide a read-out of the monkey's categorical decision. Clouds represent categorical decisions. The monkey cartoon at the far right is reproduced with permission from Mendoza, G., Méndez, J.C., Pérez, O. et al. Neural basis for categorical boundaries in the primate pre-SMA during relative categorization of time intervals. Nat Commun 9, 1098 (2018). https://doi.org/10.1038/s41467-018-03482-8.

short-selective neurons synchronized with the frequency reflecting a short category, and the set of long-selective neurons synchronized with the frequency reflecting a long category. Further, Granger causality analysis revealed a beta frequency shift in the direction of dlPFC to preSMA, but not in the opposite direction. Top-down prefrontal signals have also previously been reported in the parieto-prefrontal circuit during spatial categorization in the monkey[27,28]. However the current results are based on one animal.

Biophysically, a plausible account of how beta could emerge at different frequencies is provided by Sherman and colleagues[20], who found that beta events emerge in the cortex from the integration of synchronous bursts of subthreshold excitatory synaptic input simultaneously driving proximal and distal dendrites of pyramidal neurons. If the distal input is sufficiently strong and lasts about a beta period, a beta burst can be generated. The duration of this distal drive was shown to be linearly correlated with the period of the beta burst, i.e., inversely related to its frequency. The source of the distal drive is possibly the ventromedial thalamus, known to project to supragranular layers in the prefrontal cortex[29]. This pathway has been shown to modulate the overall activity of the recipient area without eliciting spikes[30].

Prominent theoretical accounts of the function of neural oscillations propose that oscillations control the flow of signals between anatomically connected regions[31]. At the algorithmic level, beta oscillations at different frequencies could act as separate channels to selectively transmit decision information downstream, referred to in a model by Akam and Kullman as frequency-division multiplexing[32,33]. Once a neural population code encodes a decision, an oscillation at a particular frequency can serve as a channel to selectively transmit the code downstream, where a network with the appropriate filter settings can selectively read out the code[33]. Transient oscillatory bursts at distinct frequencies, as observed in our data, are particularly well-suited for this mechanism[32]. In this view, the neural population code represents the value of the signal, while the oscillatory modulation represents the metadata required to distinguish the signal from others.

Previous accounts had proposed a role for beta in maintaining mental content[16]. Recently, we proposed that beyond maintenance, beta plays a role in reactivating latent contents[15]. Here, we provide support for this account. In this experiment, the relative categories were defined at the start of each session, and this content was then reactivated during each trial's decision delay in order to correctly perform the task. With the contents likely coded at the level of neurons[11,34], we propose that beta oscillations play a role in selectively

(re-)activating the relevant neural ensembles at the right moments and selectively relaying their signals downstream (Fig. 5). Our observation that the frequency shift is consistent between the distance and duration tasks supports this view: because the downstream consequences of the decision signals are the same in both temporal and spatial versions of our task (i.e., producing behavior corresponding to "long" or "short"), the decision signals are transmitted within the same frequency channels in both task versions.

While a few empirical studies have shown frequency modulations according to task contexts in both macaques[35] and humans[36,37], we here provide the first evidence that distinct oscillatory frequencies support routing information related to distinct mental contents, i.e., categorical decisions. Our results may lead to insights into pathologies marked by a decreased capacity to categorize, such as autism[38], and highlight the importance of accounting for frequency changes in clinical treatments that rely on rhythmic stimulation[39,40]. Finally, our results imply that neurorehabilitation strategies such as BMIs controlled by beta bursts[41] could make use of distinct frequencies within the beta band.

## Methods

### Animals
Two male Rhesus monkeys (*Macaca mulatta*): monkey 1 (5.5 kg BW) and monkey 2 (7.2 kg BW), were tested. All experimental procedures were approved by the National University of Mexico Institutional Animal Care and Use Committee and conformed to the principles outlined in the Guide for Care and Use of Laboratory Animals (NIH, publication number 85–23, revised 1985).

### Materials
During the task performance, the monkeys were seated in a primate chair with their head and left arm restrained. The gaze position was measured with an infrared tracking system (ISCAN, Inc., Woburn, MA, USA). Visual stimuli were presented on a computer monitor (HP7540, 160 Hz refresh rate) 56 cm away from the monkey's eyes, using Visual Basic (Microsoft Visual Basic 6.0, 1998). The task required that monkeys manipulate a joystick (H000E-NO-C, CTI electronics, Stratford, CT, USA) to control the position of the cursor on the screen.

### Task
The details of the task have been reported previously[4,11]. Briefly, monkeys were trained to categorize the interval or distance between two visual stimuli as either "short" or "long," according to previously learned prototypes. The temporal sequence of a trial (Fig. 1a) was as follows: A circle containing a fixation point was shown in the center of the screen. The animal started the trial by staring and keeping his gaze within a circular window with a diameter of 4° of visual angle, which was centered at the fixation point, and by placing and maintaining the cursor inside the central circle. A trial started after a variable waiting period (500 + Δ 1000 ms).

In the interval categorization tasks (T1–T3), two parallel bars (8° × 0.7° of visual angle) separated by constant distance (6° of visual angle) appeared briefly (50 ms), disappeared for a particular test interval, and reappeared in the same position. The first and second stimulus presentations indicated, respectively, the beginning and the end of the test interval. While the duration of the test interval varied from trial to trial, the position of the bars was always the same. In the distance categorization tasks (S2, S2, S3), the distance between the two parallel bars varied, while the interval between the two presentations was always the same (670 ms).

After a fixed delay (500 ms for monkey 1 and 1000 ms for monkey 2−monkey 1 had lower performance levels with a delay of 1000 ms. Hence, we set the delay at 500 ms for this animal), two response targets (orange and blue circles), were presented (Fig. 1a, right). Across trials, both response targets could occupy one of eight possible

locations on the periphery of the screen, which precluded the contamination of the categorization process by preparation to move to a particular place in space. The monkeys were trained to move the cursor from the central circle to the orange target if the test interval was short or to the blue target if it was long. The monkey received a juice reward immediately after each correct response. The inter-trial interval (ITI) was 1500 ms. Eye fixation was enforced from the beginning of the trial until target presentation when monkeys were allowed to break fixation.

## Stimuli and task procedures

Six blocks of stimuli (T1–T3, S1–S3), each containing eight different intervals or distances and different between-category boundaries, were employed. In the following, we describe the procedure for the interval task; the distance task followed the same logic. The first four intervals of every block were considered "short" (Fig. 1b), and the remaining "long". Furthermore, some durations were present in two blocks but belonged to a different category in each case, emphasizing the context-dependent nature of categorization. Consequently, within recording sessions, the monkeys had to flexibly change their subjective limit between categories to successfully categorize the intervals within different blocks. All intervals of each block were presented randomly (monkey 2) or pseudo-randomly (monkey 1). For every block of stimuli, the monkey had training and a testing phase. The first 24 trials of a block of trials constituted the training phase in which only the shortest and the longest interval of each block were presented in random order (12 repetitions per interval). In this phase, the color of the stimulus bars was orange when the short interval was presented and blue for the long interval, matching the color of the correct response target and thereby defining the short and long prototypes to be memorized for this block. The following 96 trials constituted the test phase, in which every one of the eight stimuli of the current block was presented 12 times. The color of the stimulus bars during this phase was green regardless of the stimulus category, requiring the animals to remember the prototypes and/or an implicit limit or boundary interval to solve the task. In every recording session, monkey 1 performed four test blocks (T1–T3 and S2), and monkey 2 performed six test blocks (T1–T3, S1–S3) in a randomized order.

## Surgery

Recording chambers (8-mm inner diameter) were implanted over the left pre-SMA and dlPFC during aseptic surgery under Sevoflurane (1–2%) gas anesthesia. Chamber positions were determined on the basis of structural MRI (Fig. 1c). Titanium posts for head restraining were also implanted. Broad spectrum antibiotics (Enrofloxacin, 5 mg/kg/day, i.m.) and analgesics (Ketorolac 0.75 mg/kg/6 h or Tramadol 50–100 mg/4–6 h, i.m.) were administered for 3 days after surgery.

## Neuronal recordings and spike sorting

The extracellular activity of neurons in preSMA was recorded with quartz-insulated tungsten microelectrodes (1–3 MΩ) mounted in multielectrode manipulators (Eckhorn System, Thomas Recording, GMbH, Giessen, Germany). All neurons were recorded regardless of their activity during the task, and the recording site changed from session to session. Spike waveform data were sorted offline (Plexon Offline Sorter, v3.0. Plexon Inc. Dallas, TX, USA; monkey 2) or online employing window discriminators (Blackrock Microsystems LLC, Salt Lake City, UT, USA; monkey 1). LFP data were simultaneously recorded from both preSMA and dlPFC in monkey 1 and from preSMA in monkey 2, using a 250-Hz low-pass filter and stored at 1000 Hz for offline analysis.

The current dataset includes a reanalysis of the data reported by Mendoza and colleagues[11]. The dataset reported in our current study additionally includes the spatial (distance categorization) version of the task and recordings from dlPFC.

## LFP preprocessing

All LFP preprocessing and data analyses were done in Fieldtrip[42] (v. 20230118) and with custom[43] Matlab 2019b code (v. 9.7, The Mathworks Inc., Natick, Massachusetts, USA). Due to the acute nature of our recordings, where each day the electrodes needed to go through the dura, the signal-to-noise ratio in monkey 2 LFP recordings deteriorated after three recording sessions, leaving us with 18 blocks of clean preSMA data. For monkey 1, there were 199 blocks of dlPFC and preSMA data. We re-referenced the data from channels within each cortical site (dlPFC and preSMA) to the average within that site. We then visually inspected epochs starting 500 ms before the presentation of the first stimulus until reward delivery and rejected excessively noisy channels and trials. We rejected one block of dlPFC data as the entire block was noisy. From the remaining data, we rejected around 10% of noisy trials and channels.

## Spectral analysis

We computed the power spectra (2–36 Hz) for all channels during the 500-ms prestimulus interval with a fast Fourier approach and a Hanning taper, padded to 1-s length for a frequency resolution of 1 Hz. As we wanted to specifically target beta oscillations, we then selected the channel within each block and region that had the highest power in the beta frequency range (5–35 Hz) for further analysis. We analyzed the 500 ms decision interval (for monkey 2, we selected the last 500 ms of the 1000-ms delay) for those selected channels with the same approach. Finally, we averaged the single-trial power estimates within each block and cortical site, or per trial type within each block and cortical site, depending on the analysis.

To compare power differences between the prestimulus and decision delays, we used a cluster-based permutation approach[44], clustering in the spectral dimension (15–35 Hz). To compare beta peak frequencies throughout the analyses, we detected the spectral peak with the highest power in the averaged power spectra and compared conditions with paired t-tests. In addition, for this effect of peak frequency on decisions, we quantified effect size with Area Under the Receiver Operating Characteristics (AUROC). We fit a logistic regression with the averaged spectra as predictors and the decisions as response variables before computing the area under the ROC curve, using the probability estimates from the logistic regression model as scores. In cases where the compared conditions did not match (e.g., when comparing "very long" T1 vs. "very short" T2 peak frequencies), we used unpaired t-tests. When comparing "very long/short" with "less long/short" peak frequencies, we used a 2-way ANOVA with factors long/short and very/less.

## Burst analysis

To characterize the bursting nature of beta oscillations, we computed time-frequency representations (15–35 Hz in steps of 1 Hz) of the entire epoch length using a sliding time window of 250 ms in steps of 20 ms, multiplied with a Hanning taper. We then computed the mean and standard deviation of power within a block for each frequency and marked the time-frequency points that exceeded two standard deviations above the mean and lasted at least the duration of one cycle (defined as 1/frequency). We zoomed in on the prestimulus and decision intervals, and based on temporal and spectral adjacency, we clustered the marked time-frequency points into burst events. Finally, we extracted our parameters of interest from these burst events. For each trial, we counted the number of burst events and then focused on the event that contained the time-frequency point with the highest power. For each of these events, we extracted the time span, frequency span, timing relative to the interval, and peak frequency, and we computed its volume with a double integral. Finally, we averaged the trial estimates of these parameters within each block and cortical site or per trial type within each block and cortical site, depending on the analysis. We used paired t-tests to test for differences between conditions.

### Instantaneous frequency analysis

To investigate the time course of the peak beta frequency, we analyzed instantaneous frequency as detailed by Cohen (2014). Briefly, we band-passed the raw data within a range of 10 Hz around the peak beta frequency (25–35 Hz for monkey 1 decision delay), applied the Hilbert transform, extracted the phase angle time series, took its temporal derivative, applied 10 median filters, and took the median instantaneous frequency estimate across those 10 filters. We then averaged the resulting instantaneous frequency time series per trial type within each block and cortical site. We used a cluster-based permutation approach to test for differences between trial types, clustering over the 500-ms time range of the decision delay.

### Connectivity analysis

We used the Fourier coefficients obtained while computing the power spectra (described above) to estimate coherence between preSMA and dlPFC[45] and bivariate, nonparametric Granger causality[46] during the decision delay. Granger causality gave us separate estimates of the connection strengths from preSMA to dlPFC and vice versa. We computed these estimates separately for the two trial types within each block. We used peak detection and paired $t$-tests (as described in the spectral analysis section) to test for differences in peak frequencies for the coherence measure. For the Granger causality measure, we detected peaks in the same way, then used a two-way ANOVA with factors short/long and directionality.

### Cell-type identification and spike-field coherence analysis

To estimate spike-field coherence, we first obtained the spiking activity data of cells that were previously found to be category-selective[11]. Briefly, to identify these cells, we performed a multiple linear regression with the discharge rate of a cell (in a sliding window of 250 ms in steps of 25 ms) as the dependent variable and the categorical choice of monkeys, the duration or distance of the test stimulus, and the trial outcome (correct/incorrect) as factors. We calculated two additional measures that determined the association between the neural activity and the monkeys' choices for the same sliding windows: the choice-probability index[47,48], which indicates the proportion of behavioral responses that can be predicted from the neuron's activity; and a contingency table between the decoded and the observed monkey's choices across all trials, and then performed a $\chi^2$-test on this table. A cell was considered category-selective when: (1) the categorical choice factor was significant in the permutation test ($p < 0.05$) of the multiple linear regression model for at least two consecutive sliding windows during the decision delay period; (2) the choice probability index was above 0.6; and (3) the $\chi^2$-test was significant ($p < 0.05$).

After obtaining the relevant cells, we computed spike-triggered averages around their spikes and interpolated 4 ms of data around each spike ($\pm2$ ms) during the decision delay. We then computed spike-triggered spectra (2–36 Hz) for 500 ms around each spike ($\pm250$ ms) with a fast Fourier approach and a Hanning taper and used these Fourier coefficients to compute the pairwise phase consistency (PPC) as implemented in Fieldtrip[49]. The PPC is an unbiased and consistent estimate of how well two signals generated by two sources show a consistent phase relationship in a particular frequency band. To test for differences in peak PPC frequencies, we used peak detection as described above and unpaired $t$-tests.

### Inclusion and ethics statement

Our lab is committed to pursuing science in a collaborative way and contributing to a more diverse and inclusive academia. More information here: www.haegenslab.com.

### Reporting summary

Further information on research design is available in the Nature Portfolio Reporting Summary linked to this article.

## Data availability

The data generated in this study have been deposited in the OSF database under the accession code[43] https://osf.io/pza56/. The data are publicly available. Source data are provided in this paper.

## Code availability

The code used to analyze the data supporting the claims of this study is publicly available here[43] https://osf.io/pza56/.

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

## Acknowledgements

This research was funded in whole, or in part, by the Austrian Science Fund (FWF) Erwin Schrödinger Fellowship J4580 to E.R. For the purpose of open access, the author has applied a CC BY public copyright license to any Author Accepted Manuscript version arising from this submission. H.M. is supported by Consejo Nacional de Ciencia y Tecnología (CONACYT) Grant CONACYT: A1-S-8430, UNAM-DGAPA-PAPIIT IN201721. S.H. is supported by NWO Vidi 016.Vidi.185.137 and NIH R01 MH123679.

## Author contributions

E.R. analyzed the data and wrote the paper. Y.Z. helped analyze the data. G.M. performed the experiment and helped analyze the data. J.C.M. designed and performed the experiment, and edited the paper. H.M. designed and performed the experiment, and edited the paper. S.H. supervised the project and edited the paper.

## Competing interests

The authors declare no competing interests.
