## [Peer Review File · Nature Communications]

Distinct beta frequencies reflect categorical decisionsREVIEWER COMMENTS

Reviewer #1 (Remarks to the Author):

“Distinct beta frequencies reflect categorial decisions” by Rassi et al reports the finding that in dlPFC and preSMA, short vs long categories for both distance and duration are associated with high (~30Hz) vs low (~29Hz) beta frequencies. This finding is very interesting, especially the consistent effects between spatial and temporal categorization. However, there are some concerns I’d like the authors to address.

Major Comments

1. All the findings related to dlPFC, in particular Figure 4, rely on one monkey. While the granger causality analysis is interesting as an exploratory finding, one should take caution in concluding anything from it given that it came from one monkey. As such, I recommend de-emphasizing the dlPFC findings from the abstract and the text.
2. While the consistency in frequency shift between distance and duration for the representation of short vs long categories is highly interesting, it’s not clear how. Perhaps “short” category is represented with a smaller ensemble, and synchronizes at a faster frequency? Or maybe the neurons representing the “short” category has a higher firing rate during the decision delay period and this leads to higher beta frequency? Some discussion on plausible biophysical mechanisms would be helpful and interesting to the readers.

Medium-level Comments

1. The experimental design is very elegant in that the same exact interval and distance values could be short in one block and long in another block. However, there is no direct comparison of these overlapping values in different blocks. It would be very powerful to show the frequency effects for identical stimuli in different blocks (e.g., 450 or 500ms trials for T1 block vs T2 block).
2. I recommend including Table 5 and Table 7 equivalents for each monkey. Even if the frequency shift effects are not significant for both monkeys, it would be important to confirm that the effects are in the same direction.
3. With the p-values and t-values, it is difficult to infer the effect size. Some measure of effect size (e.g., AUROC when using frequency to predict monkey’s decision) would be helpful.
4. In Figure 4, spike-field coherence panel for preSMA is missing?

Minor Comments

1. In the spatial task, was the first and second stimulus within a trial identical? Please clarify.
2. In the Introduction, it would be helpful to mention why the authors chose to record from dIPFC and preSMA during this categorization task.

Reviewer #2 (Remarks to the Author):

This study presents an analysis of electrophysiological recordings from macaque pre-SMA and dIPFC during two magnitude categorization tasks (one based on temporal interval, the other on spatial intervals). The authors compute frequency spectra of local LFP power, inter-area LFP coherence, and within-area spike-field coherence between and compare them between long vs. short category trials. They also compare spectra of inter-area Granger causality, in both directions (preSMA → dIPFC and vice versa). The main finding is a peak shift between long vs. short judgments for the spectra of all the above measures, specifically within the beta frequency range (the most prominent peak in the overall power spectra). For LFP power, this peak shift also reflects the reported judgment rather than the stimulus (i.e., flipping with respect to stimulus category for error trials) and is categorical in nature, rather than reflecting the continuous differences present in the stimuli.

I found the reported effects to be interesting, and the methods are sound. At the same time, the study seems to have conceptual shortcomings and the analyses presented seems to be somewhat incomplete.

1. The study falls short of providing an account for how the peak frequency of LFP oscillations should encode categorical decisions about magnitude. How does this peak shift come about, and how is it read out downstream to drive behavioral choice? A simple model could help here.
2. Relatedly, the study by Mendoza et al, Nat Commun (2018) provided a beautiful and careful analysis of the encoding of stimulus categories and animals' decisions in the firing rate dynamics of individual preSMA neurons in these tasks. The current paper would be a lot more impactful if it related the current LFP analyses more directly to those of the spiking activity, in terms of their functional properties (which information is present in spiking that is not present in the LFP spectra) as well as in terms of their correlation (i.e., spike-field coherence, currently only shown for dIPFC, not preSMA).
3. The authors claim that the results are the same for the temporal and spatial tasks, but all spectra shown in the main figures are collapsed across both tasks. There seems to be some information about the estimated peak parameters for the individual tasks in the supplementary tables (unclear, due to use

of shorthands for the conditions which are not defined), but this is not sufficient to evaluate the effects. Please show separate spectra for the two tasks.

4. It is unclear whether the inter-area coherence and Granger causality effects reflect the stimulus or the categorical judgments. The error analyses should also be done for these measures.

5. The authors claim that the peak frequency shift emerges from the beta frequency drive from dLPFC to preSMA. But the peak shifts are visible in the spectra for both directions, they are just not statistically significant for the direction preSMA -> dIPFC. To make a stronger case about the origin of the effect, the authors should quantitatively compare the beta peak shifts between the two directions (dIPFC -> preSMA -vs. preSMA -> dIPFC).

Minor:

Is this a re-analysis of the data from Mendoza et al, Nat Commun (2018)? I haven't found an explicit statement about this anywhere in Methods. I applaud efforts to re-analyze such datasets for multiple independent studies. But it is important to be clear and transparent about this.

Response to Reviewer 1

Major Comments

1. All the findings related to dlPFC, in particular Figure 4, rely on one monkey. While the granger causality analysis is interesting as an exploratory finding, one should take caution in concluding anything from it given that it came from one monkey. As such, I recommend de-emphasizing the dlPFC findings from the abstract and the text.

We agree with the reviewer that caution is warranted when it comes to interpreting the findings related to dlPFC, as they rely on one monkey. Unfortunately, dlPFC data was not available from the second monkey. As such, we have now de-emphasized these findings in the abstract, results, and discussion sections. We have additionally clarified in the results and discussion sections that since dlPFC data was only available for one monkey, those findings should be considered tentative and not conclusive.

Results, page 7 *“Note that these simultaneous recordings were only available in one animal, so the following results should be treated with caution.”*

page 8: *“However, we cannot draw strong conclusions from the above results relating to connectivity and spike-field coherence, as these results rely on data from one animal.”*

Discussion, page 12: *“Top-down prefrontal signals have also previously been reported in the parieto-prefrontal circuit during spatial categorization in the monkey^{27,28}; however the current results are based on one animal.”*

2. While the consistency in frequency shift between distance and duration for the representation of short vs long categories is highly interesting, it’s not clear how. Perhaps “short” category is represented with a smaller ensemble, and synchronizes at a faster frequency? Or maybe the neurons representing the “short” category has a higher firing rate during the decision delay period and this leads to higher beta frequency? Some discussion on plausible biophysical mechanisms would be helpful and interesting to the readers.

We thank the reviewer for thinking along about the plausible biophysical mechanisms that lead to the frequency shift. We were not able to verify that “short” category is represented with a smaller ensemble, as there were on average more short-selective cells than long-selective cells in our sample. We also observed no differences in the average firing rates of short- and long-selective cells. We now report these observations descriptively in the text but do not draw conclusions from them. Instead, we offer a plausible biophysical mechanism for how the frequency shift is generated based on the modelling work in Sherman et al. (2016, PNAS), and a plausible algorithmic mechanism of how it is transmitted downstream based on the modelling work in Akam & Kullman (2014, Nat Rev Neurosci). Finally, we speculate that the consistency in frequency shift between distance and duration tasks is likely because the downstream consequence of the frequency shift is the same in both cases (i.e., producing behavior corresponding to “long” or “short” categorization).

Results, page 8:

“[...] we calculated spike-field coherence separately for short-selective neurons (preSMA n=478, dlPFC n=445) with “short” trials, and long-selective neurons (preSMA n=391, dlPFC n=329) with “long” trials.

Note that there were no significant differences in the firing rates of these “short” vs “long” cells (both $t < 1$, both $p > .3$).”

Discussion, page 12:

“Biophysically, a plausible account of how beta could emerge at different frequencies is provided by Sherman and colleagues²⁰, who found that beta events emerge in cortex when synchronous bursts of subthreshold excitatory synaptic input are simultaneously integrated by proximal and distal dendrites of pyramidal neurons. If the distal input is sufficiently strong and lasts about a beta period, a beta burst can be generated. The duration of this distal drive was shown to be linearly correlated with the period of the beta burst, so inversely related to its frequency. The source of the distal drive is possibly the ventromedial thalamus, known to project to supragranular layers in prefrontal cortex²⁹. This pathway has been shown to modulate the overall activity of the recipient area without eliciting spikes³⁰.

Prominent theoretical accounts of the function of neural oscillations propose that oscillations control the flow of signals between anatomically connected regions³¹. At the algorithmic level, beta oscillations at different frequencies could act as separate channels to selectively transmit decision information downstream, referred to in a model by Akam and Kullman as frequency-division multiplexing^{32,33}. Once a neural population code encodes a decision, an oscillation at a particular frequency can serve as a channel to selectively transmit the code downstream, where a network with the appropriate filter settings can selectively read out the code³³. Transient oscillatory bursts at distinct frequencies, as observed in our data, are particularly well-suited for this mechanism³². In this view, the neural population code represents the value of the signal, while the oscillatory modulation represents the metadata required to distinguish the signal from others.

Previous accounts had proposed a role for beta in maintaining mental content¹⁶. Recently, we proposed that beyond maintenance, beta plays a role in reactivating latent contents¹⁵. Here, we provide support for this account. In this experiment, the relative categories were defined at the start of each session, and this content was then reactivated during each trial’s decision delay in order to correctly perform the task. With the contents likely coded at the level of neurons^{11,34}, we propose that beta oscillations play a role in selectively (re-)activating the relevant neural ensembles at the right moments, and selectively relaying their signals downstream. Our observation that the frequency shift is consistent between the distance and duration tasks supports this view: because the downstream consequences of the decision signals are the same in both temporal and spatial versions of our task (i.e., producing behavior corresponding to “long” or “short”), the decision signals are transmitted within the same frequency channels in both task versions.”

11. Mendoza, G., Méndez, J. C., Pérez, O., Prado, L. & Merchant, H. Neural basis for categorical boundaries in the primate pre-SMA during relative categorization of time intervals. *Nat Commun* **9**, 1098 (2018).
15. Spitzer, B. & Haegens, S. Beyond the Status Quo: A Role for Beta Oscillations in Endogenous Content (Re)Activation. *eNeuro* **4**, (2017).
16. Engel, A. K. & Fries, P. Beta-band oscillations—signalling the status quo? *Current Opinion in Neurobiology* **20**, 156–165 (2010).

20. Sherman, M. A. *et al.* Neural mechanisms of transient neocortical beta rhythms: Converging evidence from humans, computational modeling, monkeys, and mice. *PNAS* **113**, E4885–E4894 (2016).
29. Herkenham, M. Laminar organization of thalamic projections to the rat neocortex. *Science* **207**, 532–535 (1980).
30. Reichova, I. & Sherman, S. M. Somatosensory Corticothalamic Projections: Distinguishing Drivers From Modulators. *Journal of Neurophysiology* **92**, 2185–2197 (2004).
31. Fries, P. Rhythms For Cognition: Communication Through Coherence. *Neuron* **88**, 220–235 (2015).
32. Akam, T. & Kullmann, D. M. Oscillatory multiplexing of population codes for selective communication in the mammalian brain. *Nat Rev Neurosci* **15**, 111–122 (2014).
33. Akam, T. & Kullmann, D. M. Oscillations and filtering networks support flexible routing of information. *Neuron* **67**, 308–320 (2010).
34. Kreiman, G., Koch, C. & Fried, I. Category-specific visual responses of single neurons in the human medial temporal lobe. *Nat. Neurosci.* **3**, 946–953 (2000).

Medium-level Comments

1. The experimental design is very elegant in that the same exact interval and distance values could be short in one block and long in another block. However, there is no direct comparison of these overlapping values in different blocks. It would be very powerful to show the frequency effects for identical stimuli in different blocks (e.g., 450 or 500ms trials for T1 block vs T2 block).

We apologize that the contrast for identical stimuli in different blocks was perhaps not sufficiently highlighted in the text. We had reported this in the results section “Two distinct beta-band frequencies reflected the two relative categorical decisions across tasks and recording sites”, (page 6, third paragraph), referring to the relevant stimuli as “very long” and “very short”. We reported this finding for both T1 vs T2 and T2 vs T3; however, in re-reading the section, we realized there were typos in that paragraph which might have made the section unclear. We have now corrected the typos. Note that in the main text, we pooled across the two longest stimuli in T1 and the two shortest in T2 (i.e., 450ms and 500ms) and did the same for the T2 and T3, resulting in two contrasts per region. We have now additionally included the contrasts for each individual overlapping stimulus (i.e., 450ms and 500ms separately) as Tables 9 (dIPFC) and 10 (preSMA).

Monkey 1 dIPFC	long trials (Hz) mean (+/- SD)	short trials (Hz) mean (+/- SD)	short > long (t, df)	short > long (p)
450ms	29.1 (+/- 1.9)	30.2 (+/- 2.1)	2.6, 99	.011
500ms	29.1 (+/- 1.6)	30.1 (+/- 2.0)	2.9, 99	.005

870ms	28.5 (+/- 1.6)	30.1 (+/- 1.7)	4.4, 81	4e-5
920ms	28.9 (+/- 2.0)	30.5 (+/- 1.5)	6.42, 82	8e-9

Table 9. Beta frequency shift in dIPFC during decision delay of the duration categorization tasks, for trials with identical stimuli within different task versions (i.e., trials with identical stimuli categorized as “short” in one task version but “long” in another; see overlapping stimuli outlined in Figure 1b).

Monkey 1 preSMA	long trials (Hz) mean (+/- SD)	short trials (Hz) mean (+/- SD)	short > long (t, df)	short > long (p)
450ms	29.2 (+/- 2.5)	30.1 (+/- 2.1)	1.99, 99	.049
500ms	28.9 (+/- 2.0)	30.0 (+/- 2.3)	2.8, 99	.006
870ms	28.2 (+/- 1.6)	29.3 (+/-2.3)	2.8, 81	.007
920ms	28.4 (+/- 1.7)	29.6 (+/- 2.0)	2.9, 82	.005

Table 10. Beta frequency shift in preSMA during decision delay of the duration categorization tasks, for trials with identical stimuli within different task versions (i.e., trials with identical stimuli categorized as “short” in one task version but “long” in another; see overlapping stimuli outlined in Figure 1b).

2. I recommend including Table 5 and Table 7 equivalents for each monkey. Even if the frequency shift effects are not significant for both monkeys, it would be important to confirm that the effects are in the same direction.

Indeed, the frequency shift effect is significant for both monkeys. However, since monkey 2 data is only available for 18 sessions (~3 sessions x 6 tasks), it would not be very meaningful to split monkey 2’s data into the 6 different tasks. Instead, we have added a row to Table 5 aggregating the data from monkey 2, and removed monkey 2’s data from the remaining rows, which now represent monkey 1’s data. We’ve also added the Table 7 equivalent for monkey 2 (Table 8).

preSMA	short trials (Hz) mean (+/-SD)	long trials (Hz) mean (+/-SD)	short > long (t(df))	short > long (p)
---------------	--------------------------------	-------------------------------	----------------------	------------------

Monkey 1 - T1	30.1 (+/- 2.2)	28.1 (+/- 2.0)	5.4 (59)	1e-6
Monkey 1 - T2	29.3 (+/- 2.4)	27.7 (+/- 1.8)	4.2 (56)	1e-4
Monkey 1 - T3	29.0 (+/- 2.5)	27.4 (+/- 1.6)	3.3 (43)	.002
Monkey 1 - S2	28.7 (+/- 2.0)	27.3 (+/- 1.7)	4.7 (35)	4e-5
Monkey 2 - all tasks	25.2 (+/- 2.7)	23.6 (+/- 2.8)	2.7 (14)	.016

Table 5. Beta frequency shift in preSMA during decision delay for trials categorized as “short” vs. “long”. T1, T2, and T3 are the interval categorization tasks with the shortest, middle, and longest sets of intervals, respectively (see Figure 1b for exact values). S2 is the distance categorization task with the middle set of distances. For Monkey 2, tasks additionally included S1 and S3, the distance categorization tasks with the shortest and longest distances, respectively.

Monkey 2 preSMA	mean (+/- SD) short trials	mean (+/- SD) long trials	short > long (t)	short > long (p)
burst frequency (Hz)	25.9 (+/- 1.5)	25.2 (+/- 1.4)	3.1	.006
burst number	1.00 (+/- 0.29)	0.97 (+/- 0.41)	-0.5	.594
burst peak power (a.u.)	4.68 (+/- 0.71)	4.59 (+/- 0.74)	0.8	.436
burst time span (ms)	114 (+/- 15)	128 (+/- 31)	-2.0	.061
burst frequency span (Hz)	3.93 (+/- 1.23)	4.37 (+/- 1.22)	-2.6	.018

Table 8. Beta burst profile in Monkey 2 preSMA during decision delay for trials categorized as “short” vs. “long”. All t-value df = 17.

3. With the p-values and t-values, it is difficult to infer the effect size. Some measure of effect size (e.g., AUROC when using frequency to predict monkey’s decision) would be helpful.

This is a good suggestion. Using frequency to predict monkeys’ decisions, we obtained AUROC of .751 for monkey 1 dIPFC, .724 for monkey 1 preSMA, and .660 for monkey 2 preSMA.

Note that these values do not represent predictions for monkeys’ decisions on a single-trial basis. Because frequency spectra at the single-trial level are very noisy, our unit of observation for statistics throughout the manuscript was at the session level. We applied the same logic here, averaging the spectra separately for each decision within a session, before fitting a logistic regression with frequency as predictor.

We have now added the AUROC values to the main text (Results, pages 5 and 6) , and explained how we obtained these values in the Methods section (Spectral analysis, page 15):

“In addition, for this effect of peak frequency on decisions, we quantified effect size with Area Under the Receiver Operating Characteristics (AUROC). We fit a logistic regression with the averaged spectra as predictors and the decisions as response variables, before computing the area under the ROC curve, using the probability estimates from the logistic regression model as scores.”

4. In Figure 4, spike-field coherence panel for preSMA is missing?

We had originally chosen not to show the preSMA data as there was no clear pattern in those data. We have now included a figure of spike-field coherence in preSMA for completeness. We also corrected an error in the way we had plotted the shaded error regions of the dIPFC spike-field coherence data.

Figure S5. Spike-field coherence during the decision delay. **(a)** Spike-field coherence in dIPFC between short-selective neurons and the LFP during short-categorized trials, and between long-selective neurons and the LFP during long-categorized trials: the peak frequency reflected the categorical decision. **(b)** Same for preSMA: there were no significant peaks in the beta range.

Minor Comments

1. In the spatial task, was the first and second stimulus within a trial identical? Please clarify.

Yes, in both task versions, the first and second stimuli were identical within a trial. This is now clarified in the first paragraph of the Results section (page 3).

“Although the distance between bars varied between trials in the spatial task, it was identical within a trial in both task versions.”

2. In the Introduction, it would be helpful to mention why the authors chose to record from dIPFC and preSMA during this categorization task.

Thank you for the suggestion. We have added the following sentences in the introduction (page 2):

“dIPFC is known to play a central role in categorization¹² and is part of the magnitude system for time, space, and quantity¹³. It is deeply connected with preSMA, which is known to be a major node in the time processing network¹⁴, and contains cells that have been shown to encode the boundary between categories¹¹.”

11. Mendoza, G., Méndez, J. C., Pérez, O., Prado, L. & Merchant, H. Neural basis for categorical boundaries in the primate pre-SMA during relative categorization of time intervals. *Nat Commun* **9**, 1098 (2018).

12. Miller, E. K., Freedman, D. J. & Wallis, J. D. The prefrontal cortex: categories, concepts and cognition. *Philos Trans R Soc Lond B Biol Sci* **357**, 1123–1136 (2002).

13. Walsh, V. A theory of magnitude: common cortical metrics of time, space and quantity. *Trends Cogn Sci* **7**, 483–488 (2003).

14. Merchant, H., Harrington, D. L. & Meck, W. H. Neural Basis of the Perception and Estimation of Time. *Annual Review of Neuroscience* **36**, 313–336 (2013).

Response to Reviewer 2

Major

1. The study falls short of providing an account for how the peak frequency of LFP oscillations should encode categorical decisions about magnitude. How does this peak shift come about, and how is it read out downstream to drive behavioral choice? A simple model could help here.

We agree with the reviewer that more discussion of the relationship between peak frequency and categorical decisions is warranted. We would like to first clarify that we do not claim that the LFP encodes decisions. Most likely the categorical decisions are encoded (in spike firing patterns) by boundary cells and category-selective cells as elegantly shown by Mendoza et al., 2018. What we observe in the LFP peak frequency is a signal that reflects the decision, likely a channel to transmit the decision downstream. We now offer a plausible biophysical mechanism for how the frequency shift is generated based on the modelling work in Sherman et al. (2016, PNAS), and a plausible algorithmic mechanism of how it is transmitted downstream based on the modelling work in Akam & Kullman (2014, Nat Rev Neurosci).

Discussion, page 12:

“Biophysically, a plausible account of how beta could emerge at different frequencies is provided by Sherman and colleagues²⁰, who found that beta events emerge in cortex when synchronous bursts of subthreshold excitatory synaptic input are simultaneously integrated by proximal and distal dendrites of pyramidal neurons. If the distal input is sufficiently strong and lasts about a beta period, a beta burst can be generated. The duration of this distal drive was shown to be linearly correlated with the period of the beta burst, so inversely related to its frequency. The source of the distal drive is possibly the ventromedial thalamus, known to project to supragranular layers in prefrontal cortex²⁹. This pathway has been shown to modulate the overall activity of the recipient area without eliciting spikes³⁰.

Prominent theoretical accounts of the function of neural oscillations propose that oscillations control the flow of signals between anatomically connected regions³¹. At the algorithmic level, beta oscillations at different frequencies could act as separate channels to selectively transmit decision information downstream, referred to in a model by Akam and Kullman as frequency-division multiplexing^{32,33}. Once a neural population code encodes a decision, an oscillation at a particular frequency can serve as a channel to selectively transmit the code downstream, where a network with the appropriate filter settings can selectively read out the code³³. Transient oscillatory bursts at distinct frequencies, as observed in our data, are particularly well-suited for this mechanism³². In this view, the neural population code represents the value of the signal, while the oscillatory modulation represents the metadata required to distinguish the signal from others.

Previous accounts had proposed a role for beta in maintaining mental content¹⁶. Recently, we proposed that beyond maintenance, beta plays a role in reactivating latent contents¹⁵. Here, we provide support for this account. In this experiment, the relative categories were defined at the start of each session, and this content was then reactivated during each trial’s decision delay in order to correctly perform the task. With the contents likely coded at the level of neurons^{11,34}, we propose that beta oscillations play a role in selectively (re-)activating the relevant neural ensembles at the right moments, and selectively relaying their signals downstream. Our observation that the frequency shift is consistent between the distance and duration tasks supports this view: because the downstream consequences of the decision signals are the same in both temporal and spatial versions of our task (i.e., producing behavior

corresponding to “long” or “short”), the decision signals are transmitted within the same frequency channels in both task versions.”

11. Mendoza, G., Méndez, J. C., Pérez, O., Prado, L. & Merchant, H. Neural basis for categorical boundaries in the primate pre-SMA during relative categorization of time intervals. *Nat Commun* **9**, 1098 (2018).
15. Spitzer, B. & Haegens, S. Beyond the Status Quo: A Role for Beta Oscillations in Endogenous Content (Re)Activation. *eNeuro* **4**, (2017).
16. Engel, A. K. & Fries, P. Beta-band oscillations—signalling the status quo? *Current Opinion in Neurobiology* **20**, 156–165 (2010).
20. Sherman, M. A. *et al.* Neural mechanisms of transient neocortical beta rhythms: Converging evidence from humans, computational modeling, monkeys, and mice. *PNAS* **113**, E4885–E4894 (2016).
29. Herkenham, M. Laminar organization of thalamic projections to the rat neocortex. *Science* **207**, 532–535 (1980).
30. Reichova, I. & Sherman, S. M. Somatosensory Corticothalamic Projections: Distinguishing Drivers From Modulators. *Journal of Neurophysiology* **92**, 2185–2197 (2004).
31. Fries, P. Rhythms For Cognition: Communication Through Coherence. *Neuron* **88**, 220–235 (2015).
32. Akam, T. & Kullmann, D. M. Oscillatory multiplexing of population codes for selective communication in the mammalian brain. *Nat Rev Neurosci* **15**, 111–122 (2014).
33. Akam, T. & Kullmann, D. M. Oscillations and filtering networks support flexible routing of information. *Neuron* **67**, 308–320 (2010).
34. Kreiman, G., Koch, C. & Fried, I. Category-specific visual responses of single neurons in the human medial temporal lobe. *Nat. Neurosci.* **3**, 946–953 (2000).

2. Relatedly, the study by Mendoza et al, Nat Commun (2018) provided a beautiful and careful analysis of the encoding of stimulus categories and animals' decisions in the firing rate dynamics of individual preSMA neurons in these tasks. The current paper would be a lot more impactful if it related the current LFP analyses more directly to those of the spiking activity, in terms of their functional properties (which information is present in spiking that is not present in the LFP spectra) as well as in terms of their correlation (i.e., spike-field coherence, currently only shown for dIPFC, not preSMA).

We thank the reviewer for helping us elaborate on points that will make our paper more impactful. We think our response to the previous comment addresses the current comment regarding the functional properties of spikes and LFP. We would like to add that the LFP provides a network-level source of information that can be translated to non-invasively recorded signals (EEG or MEG). There is no non-invasively recorded signal that corresponds to spiking activity. In that sense, experiments on healthy human populations can more directly benefit from insights gained by analyzing LFP spectra.

Regarding spike-field coherence, we had originally chosen not to show the preSMA data as there was no clear pattern in those data. We have now included a figure of spike-field coherence in preSMA for

completeness. We also corrected an error in the way we had plotted the shaded error regions of the dIPFC spike-field coherence data.

Figure S5. Spike-field coherence during the decision delay. **(a)** Spike-field coherence in dIPFC between short-selective neurons and the LFP during short-categorized trials, and between long-selective neurons and the LFP during long-categorized trials: the peak frequency reflected the categorical decision. **(b)** Same for preSMA: there were no significant peaks in the beta range.

3. The authors claim that the results are the same for the temporal and spatial tasks, but all spectra shown in the main figures are collapsed across both tasks. There seems to be some information about the estimated peak parameters for the individual tasks in the supplementary tables (unclear, due to use of shorthands for the conditions which are not defined), but this is not sufficient to evaluate the effects. Please show separate spectra for the two tasks.

We apologize for the lack of clarity in showing the data from the individual tasks. We have now clarified our usage of the shorthand terms in the table captions (T1, T2, T3 are the interval categorization tasks with the shortest to longest intervals respectively; similarly, S1, S2, S3 are the distance categorization tasks with shortest to longest distances respectively). We now have also included figures showing the spectra for the separate tasks.

Figure S1. Beta peak frequency in monkey 1 dIPFC reflected the categorical decision during the decision delay in each version of the task. Power spectra for “long” stimulus (blue) vs. “short” stimulus trials (orange) during trials with correct responses (**a-c**) in the temporal categorization versions of the task with the shortest to longest stimuli respectively, and (**d**) the distance categorization task (see Figure 1b for exact values and Table 4 for statistics). Shaded regions around the line graphs represent the standard error of the mean.

Figure S2. Beta peak frequency in monkey 1 preSMA reflected the categorical decision during the decision delay in each version of the task. Power spectra for “long” stimulus (blue) vs. “short” stimulus trials (orange) during trials with correct responses (**a-c**) in the temporal categorization versions of the task with the shortest to longest stimuli respectively, and (**d**) the distance categorization task (see Figure 1b for exact values and Table 5 for statistics). Shaded regions around the line graphs represent the standard error of the mean.

Figure S3. Beta peak frequency in monkey 2 preSMA reflected the categorical decision during the decision delay in each version of the task. Power spectra for “long” stimulus (blue) vs. “short” stimulus trials (orange) during trials with correct responses **(a)** in the temporal categorization versions of the task (pooled together) and **(b)** the distance categorization versions of the task (pooled together; see Figure 1b for exact stimulus values and Table 5 for statistics). Shaded regions around the line graphs represent the standard error of the mean.

4. It is unclear whether the inter-area coherence and Granger causality effects reflect the stimulus or the categorical judgments. The error analyses should also be done for these measures.

We thank the reviewer for this suggestion. Coherence and Granger causality effects indeed reflected the categorical judgement and not the stimuli: the direction of the frequency shift was reversed on error trials. We now report this analysis in the Results section (Inter-areal connectivity and spike-field coherence at the same distinct frequencies also reflected the relative categories, pages 7 and 8), and include the figures below representing this analysis in the supplementary materials.

“On incorrectly categorized trials, the frequency shift was reversed, such that coherence on “long” trials incorrectly categorized as “short” peaked at a higher frequency than “short” trials incorrectly categorized as “long” (Figure S4a; $t(163)=-7.6$, $p=2.5e-12$).”

“On incorrectly categorized trials, the frequency shift was reversed, such that dIPFC to preSMA Granger causality on “long” trials incorrectly categorized as “short” peaked at a higher frequency than “short” trials incorrectly categorized as “long” (Figure S4b; $t(169)=-3.8$, $p=2e-4$). In the direction of preSMA to dIPFC, there were no differences in Granger causality peak frequencies between the two trial types on incorrectly categorized trials (Figure S4c; $t(169)=-0.7$, $p=.50$, ns).”

Please note that this result has been de-emphasized in the manuscript as suggested by Reviewer 1, as it depends on data from one monkey.

Figure S4. Peak frequencies of between-region connectivity reflected the categorical decision during the decision delay, even when the decision was incorrect. **(a)** Coherence between preSMA and dIPFC on incorrect trials. The direction of the peak frequency shift was reversed compared with correct trials (see Figure 4a). **(b, c)** Granger causality between preSMA and dIPFC. The direction of the peak frequency of dIPFC to preSMA and that of preSMA to dIPFC Granger causality were reversed compared with correct trials (see Figure 4 c, d). However, on both correct and incorrect trials, peak frequency of dIPFC to preSMA, but not preSMA to dIPFC Granger causality, reflected the categorical decision. Shaded regions around the line graphs represent the standard error of the mean.

5. The authors claim that the peak frequency shift emerges from the beta frequency drive from dIPFC to preSMA. But the peak shifts are visible in the spectra for both directions, they are just not statistically significant for the direction preSMA -> dIPFC. To make a stronger case about the origin of the effect, the authors should quantitatively compare the beta peak shifts between the two directions (dIPFC -> preSMA -vs. preSMA -> dIPFC).

We thank the reviewer for this suggestion. We have now done this direct comparison and added it to the text, in the Results section (Inter-areal connectivity and spike-field coherence at the same distinct frequencies also reflected the relative categories), page 8:

“ difference in peak shifts between the two directions [dIPFC -> preSMA vs. preSMA -> dIPFC]: $t(183)=2.9$, $p=.004$ “

Please note that this result has been de-emphasized in the manuscript as suggested by Reviewer 1, as it depends on data from one monkey.

Minor:

Is this a re-analysis of the data from Mendoza et al, Nat Commun (2018)? I haven't found an explicit statement about this anywhere in Methods. I applaud efforts to re-analyze such datasets for multiple independent studies. But it is important to be clear and transparent about this.

Yes, parts of the current dataset were analyzed in Mendoza et al. 2018. The current dataset additionally includes the spatial (distance categorization) version of the task, and recordings from dIPFC. We have now included an explicit statement about this in Methods, "Neuronal recordings and spike sorting":

"The current dataset includes a reanalysis of the data reported by Mendoza and colleagues¹¹. The dataset reported in our current study additionally includes the spatial (distance categorization) version of the task, and recordings from dIPFC."

11. Mendoza, G., Méndez, J. C., Pérez, O., Prado, L. & Merchant, H. Neural basis for categorical boundaries in the primate pre-SMA during relative categorization of time intervals. *Nat Commun* **9**, 1098 (2018).

REVIEWERS' COMMENTS

Reviewer #1 (Remarks to the Author):

The paper entitled “Distinct beta frequencies reflect categorical decisions” reports beta oscillation frequency as a novel neural correlate of categorial decisions, Rassi et al. show that in dorsolateral prefrontal cortex (dlPFC) and in pre-supplementary motor area (preSMA) of the monkey, high vs low beta frequencies reflect short vs long categories in both duration and distance. Notably, the frequencies reflected the category relevant for the categorization task, rather than the absolute value of the duration or distance. This finding suggests that beta frequency may serve as a channel for cortico-cortical communication between dlPFC and preSMA, carrying information about categorial decisions.

The revised manuscript has thoroughly addressed all of my prior concerns. I recommend this manuscript for publication.

Reviewer #2 (Remarks to the Author):

The reviewers have done a good job in addressing my concerns.